



# The ability of a stochastic regional weather generator to reproduce heavy precipitation events across scales

Xiaoxiang Guan[1], Dung Viet Nguyen[1], Paul Voit[2], Bruno Merz[1,2], Maik Heistermann[2], Sergiy Vorogushyn[1]

[1]GFZ German Research Centre for Geosciences, Section Hydrology, Potsdam, 14473, Germany
[2]Institute of Environmental Science and Geography, University of Potsdam, Potsdam, 14469, Germany

*Correspondence to*: Xiaoxiang Guan (guan@gfz-potsdam.de)

**Abstract.** We assess the ability of a regional weather generator to represent the extremity of heavy precipitation events (HPEs) across spatial and temporal scales. To this end, we implement the multi-site non-stationary Regional Weather Generator (nsRWG) for the area of Germany and generate 100 sets of synthetic daily precipitation data spanning 72 years. The weather extremity index (WEI) and its recent cross-scale modification (xWEI) are applied to quantify the cross-scale extremity of synthetic and observed HPEs and to compare their distributions. The results show that the nsRWG excels in replicating the extremity patterns for almost all 7 durations (ranging from 1 to 7 days) considered. The frequency of small-scale 1-day rainfalls is however slightly overestimated. nsRWG aptly reproduces the potential influential areas of HPEs, whether of short or long duration. It is capable of generating precipitation events mirroring the extremity patterns observed during past disaster-causing HPEs in Germany, while simultaneously accommodating their variations. This study demonstrates the potential of the nsRWG for simulating HPE-related hazard and assessing flood risks.

## 1 Introduction

Heavy precipitation events (HPEs) are rare weather phenomena which accumulate exceptional amounts of rainfall within hours to days, over areas ranging from a few to tens of thousands of square kilometers. As the main cause of damaging floods and landslides, HPEs are the costliest natural disasters in Europe (Gvoždíková et al., 2019; NatCatSERVICE, 2023; Banerjee et al., 2024). Climate change is expected to increase the frequency, intensity and spatial extent of HPEs (Christensen and Christensen, 2003; Lenderink and Fowler, 2017; Matte et al., 2022; Yang et al., 2023) and their associated impacts (Merz et al., 2021).

Germany has experienced several notable HPEs in recent years (Hu and Franzke, 2020). For example, in August 2002 heavy precipitation led to record-high water levels in the Elbe River and its tributaries (Kreibich, et al., 2017; Thieken et al., 2022). In July 2021, a widespread HPE hit the western and southern parts of Germany as well as neighboring countries and caused one of the most devastating flood events in German history with more than 190 fatalities and €33 billion damages (Apel et al., 2022, Szönyi et al., 2022, Mohr et al., 2023).



Stochastic weather generators (WGs) can be instrumental in generating synthetic HPEs, thereby supporting flood risk management and climate adaptation (Breinl et al., 2013; Chen and Brissette, 2014; Harris et al., 2014). WGs generate a sequence of synthetic weather variables, such as precipitation and air temperature, for a specific location. WGs can be used in conjunction with various models to better understand and prepare for HPEs and their consequences when historical records or projected future time series are limited (Mehrotra and Sharma, 2010; Zhou et al., 2020). For instance, long-term synthetic weather fields (like precipitation and air temperature) generated by WGs can be used as meteorological forcing for hydrological models in order to quantify HPE-related floods and damages (Apel et al., 2016; Qin and Lu, 2014; Sairam et al., 2021). This approach allows to develop exceptional flood events needed for flood design or disaster management planning, as the generation of very long time series of flood flows and inundation increases the probability of obtaining situations where the unfortunate superposition of the different flood-generating processes leads to severe impacts (Falter et al., 2015). Such situations are rarely encountered in measured time series that are usually very limited in length.

A large number of WG models have been introduced so far, based on various statistical methods (for a review, see Maraun et al., 2010, Haberlandt et al., 2011, Serinaldi and Kilsby, 2014, Benoit and Mariethoz, 2017). Evaluating the performance of a WG is crucial to ensure that it accurately represents historical weather data and produces synthetic weather sequences that are adequate for the application context, e.g., flood estimation, drought assessment, climate change impact assessment (Tseng et al., 2020). The evaluation process helps to identify biases or limitations in the model output and fosters model improvements (e.g., Breinl et al., 2013, Serinaldi and Kilsby, 2014, Nguyen et al., 2021). Widely used metrics to evaluate synthetic precipitation data include mean, standard deviation and skewness, lag-1 autocorrelation, frequency of wet (or dry) days, and precipitation intermittency (Steinschneider and Brown, 2013; Tseng et al., 2020; Zhou et al., 2020; Nguyen et al., 2021). Additionally, the performance of WGs in simulating the extremity of precipitation events is of special interest. The extremity of precipitation events is usually described by intensity, duration and spatial extent statistics (Beniston and Stephenson, 2004; Jeferson de Medeiros et al., 2022; Müller and Kaspar, 2014; Zhang et al., 2011). The spatial consistency of multi-site WGs is sometimes evaluated based on the areal mean of the synthetic field within a region, e.g., catchment average rainfall (Ullrich et al., 2021). However, this method may underrepresent the variability within the affected area (Müller and Kaspar, 2014; Voit and Heistermann, 2022). Moreover, when calculated within a fixed region, the areal mean underestimates the extremity when only a part of the region is heavily affected. To address these issues, the correlation between multiple sites can be used to measure the spatial dependence structure of precipitation amount across a region (Breinl et al., 2013; Gao et al., 2021; Tseng et al., 2020). Furthermore, the continuity ratio, expressed as the ratio between the average precipitation at one location when another location is dry or wet, can capture the spatial coherence of precipitation between adjacent locations (Wilks, 1998; Breinl et al., 2013). However, both the correlation coefficient and the spatial continuity ratio are dominated by the bulk of the precipitation events rather than by the extreme values.

HPEs can vary in duration, from short, intense downpours to prolonged periods of heavy rainfall. Quantile-based thresholds of block maxima of precipitation totals, and empirical probability distribution plots are typically used to characterize the WG performance with regards to extremes (Breinl et al., 2013; Zhou et al., 2020; Nguyen et al., 2021; Ullrich et al., 2021). Such





methods are usually applied separately for different durations, neglecting the temporal coherence of HPEs which in reality can be extreme at different temporal and spatial scales simultaneously, triggering different types of flooding overlaying each other (Ramos et al., 2017). For instance, during the 2002 Elbe flood, small-scale extreme precipitation caused flash floods in several small Elbe tributaries. During the same event, extreme rainfall over large spatial scales and longer durations triggered fluvial flooding with widespread inundation and finally long-lasting groundwater flooding in the city of Dresden (Kreibich et al., 2005; Thieken et al., 2022). The ability of WGs to represent these cross-scale characteristics of precipitation events is thus essential for flood risk modeling. Up to now, no single measure has been used to evaluate the ability of WGs to capture the cross-scale extremity of rainfall.

In this study, we demonstrate a new evaluation approach for a WG based on the weather extremity index (WEI, Müller and Kaspar, 2014) and the cross-scale weather extremity index (xWEI) introduced by Voit and Heistermann (2022). These indices quantify the extremity of an event considering different duration levels and spatial extents. xWEI additionally integrates the extremity over different duration levels and spatial extents. Our aim is to evaluate how well the cross-scale extremity of precipitation events is captured by a WG, even if it is not specifically tailored or trained to do so.

## 2 Study area and data

The study area is Germany (Figure 1) given its exposure to hazards induced by heavy precipitation events (HPEs), as highlighted in the introduction. The non-stationary Regional Weather Generator (nsRWG) (Nguyen et al., 2024, under review) is implemented for entire Germany and parts of the upstream neighboring countries, covering the five major river catchments Elbe, Rhine, Danube, Ems and Weser, to produce long-term spatially consistent synthetic precipitation and temperature data. In this study, we focus on the ability of the WG to represent the extremity of precipitation across scales. Weather generators that comprise several catchments and cover such a large spatial area (more than 650,000 km²) are rare, and previous studies have commonly implemented WGs at smaller scales (Tseng et al., 2020; Ullrich et al., 2021; Gao et al., 2021).

Two datasets are used to parameterize nsRWG: (1) gridded precipitation data from E-OBS (version 25.0e; Cornes et al., 2018) and (2) mean sea level pressure and daily air temperature at 2 m height from the ERA5 reanalysis (Hersbach et al., 2020). Mean sea level pressure is used to classify circulation pattern types for which the local distributions of precipitation are conditioned. Regionally averaged daily temperature acts as a covariate of the local distribution in order to consider changes in precipitation for the same circulation pattern in a non-stationary warming world. Both E-OBS and ERA5 datasets are available at daily resolution, spanning from January 1950 to December 2021.

E-OBS is an ensemble gridded weather dataset and is available on a regular 0.25° grid covering Europe. It is based on station data collated by the European Climate Assessment and Dataset initiative (Cornes et al., 2018). To cope with the high computational demands, we have resampled the grid points to a reduced spatial resolution of 0.5°. The locations of the extracted E-OBS grid points are given in Figure 1.
The ERA5 dataset, provided by the European Centre for Medium-Range Weather Forecasts (ECMWF), is a state-of-the-art reanalysis dataset widely used in climate research (Hersbach et al., 2020). It provides a comprehensive and detailed representation of atmospheric conditions at the global scale. To classify the large-scale atmospheric situation, mean sea level pressure is extracted over an extent (25°N-70°N x 15°W-30°E, see Figure 1.a) that encompasses a substantial portion of Europe and adjacent regions. The regional average 2-m daily air temperature is computed for the domain 45.125°N – 55.125°N and 5.125°E – 19.125°E, which covers the nsRWG setup area.

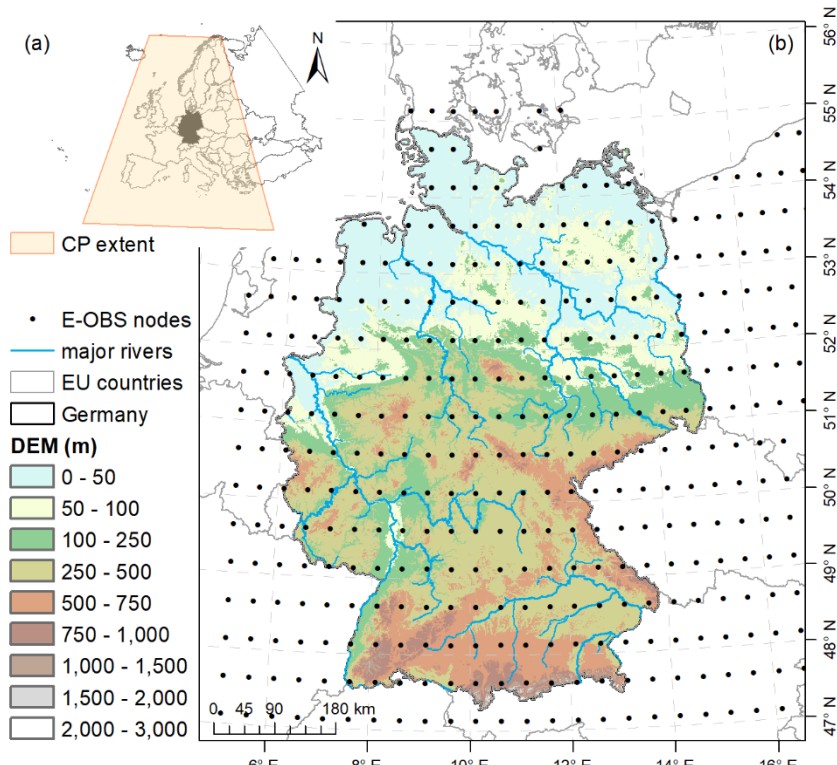

Figure 1: Study area (Germany and adjacent regions) and its topography, major rivers and E-OBS grid nodes. The extent where the mean sea level pressure was extracted to classify the circulation patterns (CP) is shown in subplot (a).

## 3 Methods

### 3.1 A stochastic multi-site non-stationary regional weather generator (nsRWG)

We adopt the non-stationary version of the multi-site regional weather generator nsRWG (Nguyen et al., 2024, under review), which is based on the original model introduced by Hundecha et al. (2012) and further refined and evaluated by Nguyen et al. (2021). Like its predecessors, nsRWG represents the spatio-temporal dependence across sites using the first-order multi-variate auto-regressive (MAR-1) model (Bardossy and Plate, 1992). The type-1 extended Generalized Pareto (extGP) distribution is used to model daily non-zero precipitation amounts. This distribution is suitable not only for





capturing both the lower bulk of precipitation amounts and extreme values but also for providing a smooth transition along the precipitation range (Naveau et al. 2016; Nguyen et al., 2021).

In the non-stationary version introduced by Nguyen et al. (2024, under review), the precipitation distribution at each site is
conditioned on the large-scale circulation pattern as a latent variable and the regional average daily temperature as a covariate of the extGP distribution scale parameter. In this way, climate variability and climate change due to changes in dynamic and thermodynamic properties of the atmosphere can be considered. Atmospheric circulation is classified into 6 circulation patterns based on mean sea level pressure for winter (November 1st - April 30th) and summer (May 1st - October 30th) seasons (12 patterns in total). We use the objective classification algorithm SANDRA (Simulated ANnealing and
Diversified RAndomization) based on the k-means clustering approach (Philipp et al., 2007). Further details about the nsRWG algorithm and configuration can be found in Nguyen et al. (2024, under review).

The cross-scale precipitation performance of nsRWG is evaluated for the E-OBS grid cells in the study area for the period from 1950-01-01 to 2021-12-31. We generate 100 realizations of synthetic precipitation datasets with a time series length of 72 years corresponding to the length of the E-OBS dataset to ensure comparability.

## 3.2 WEI and xWEI

The weather extremity index (WEI) quantifies the extremity (a product of spatial extent and rarity) of an event, as well as the spatial extent and temporal duration at which the event reached its maximum extremity (Müller and Kaspar, 2014). In this context, the spatial extent of an event is not conceived as an area of spatially contiguous grid cells, but as a number of (potentially disjoint) cells within the study region (here Germany) which exceed a certain return period. The computation of
WEI is illustrated in Figure 2. For a given spatial domain (in this case Germany), it starts with the estimation of return periods $P_{t,i}$ at each grid cell $i$, for durations from 1 to $t$ days. For each duration $t$, the grid cells in the spatial domain are sorted in decreasing order, based on their return period $P_{t,i}$ (in years), and then aggregated over increasing areas $A$ (in km$^2$) by using Eq. 1: first, $E_{tA}$ is computed for the most extreme grid cell ($n$=1). Then, the following grid cells are added to the computation of $E_{tA}$, increasing the value of $n$ by 1. For each step, the area $A$ equals the area of one single grid cell times $n$. In
the final step, $A$ corresponds to the size of the entire spatial domain. Finally, this procedure yields, for each duration $t$, a curve that shows $E_{tA}$ as a function of $A$. The value of WEI is then defined by the maximum value of $E_{tA}$ for all curves, and the spatial extent and duration at which the event was most extreme corresponds to the values of $t$ and $A$ for which this maximum of $E_{tA}$ is achieved. Note that the $E_{tA}$ curves typically have a well-defined maximum: for low values of $A$, the steep increase of $\sqrt{A/\pi}$ (the radius of a circle of size $A$) causes an increase of $E_{tA}$ with $A$. For larger values of $A$, the decrease in
return periods dominates the behavior and causes the $E_{tA}$ curve to decrease. The corresponding area where the $E_{tA}$ curve reaches its peak (WEI) is denoted here as WEI-area, which represents the spatial scale most severely affected by the HPE. The WEI-area is always much smaller than the area over which the HPE precipitation totals exceeded 0 mm. It is rather a weighted measure indicating the area in the domain which is heavily influenced by the HPE and hence prone to HPE-related impacts.





$$E_{tA} = \frac{\sum_{i=1}^{n} ln(P_{t,i})}{n} \cdot \sqrt{A/\pi} \qquad (1)$$

As each $E_{tA}$ curve represents how the extremity of an event extends across spatial scales, Voit and Heistermann (2022) proposed a cross-scale weather extremity index (xWEI) by integrating $E_{tA}$ over duration (ln(t)) and extent (*A*). xWEI quantifies how much the extremity of an event extends across both space and time (instead of the event just being extreme at one specific duration and extent). Hence, xWEI corresponds to the volume under the surface which is spanned by the $E_{tA}$ curves (Figure 2.f), placed on a grid:

$$xWEI = \iint E_{tA} \, dA \, d(ln(t)) \qquad (2)$$

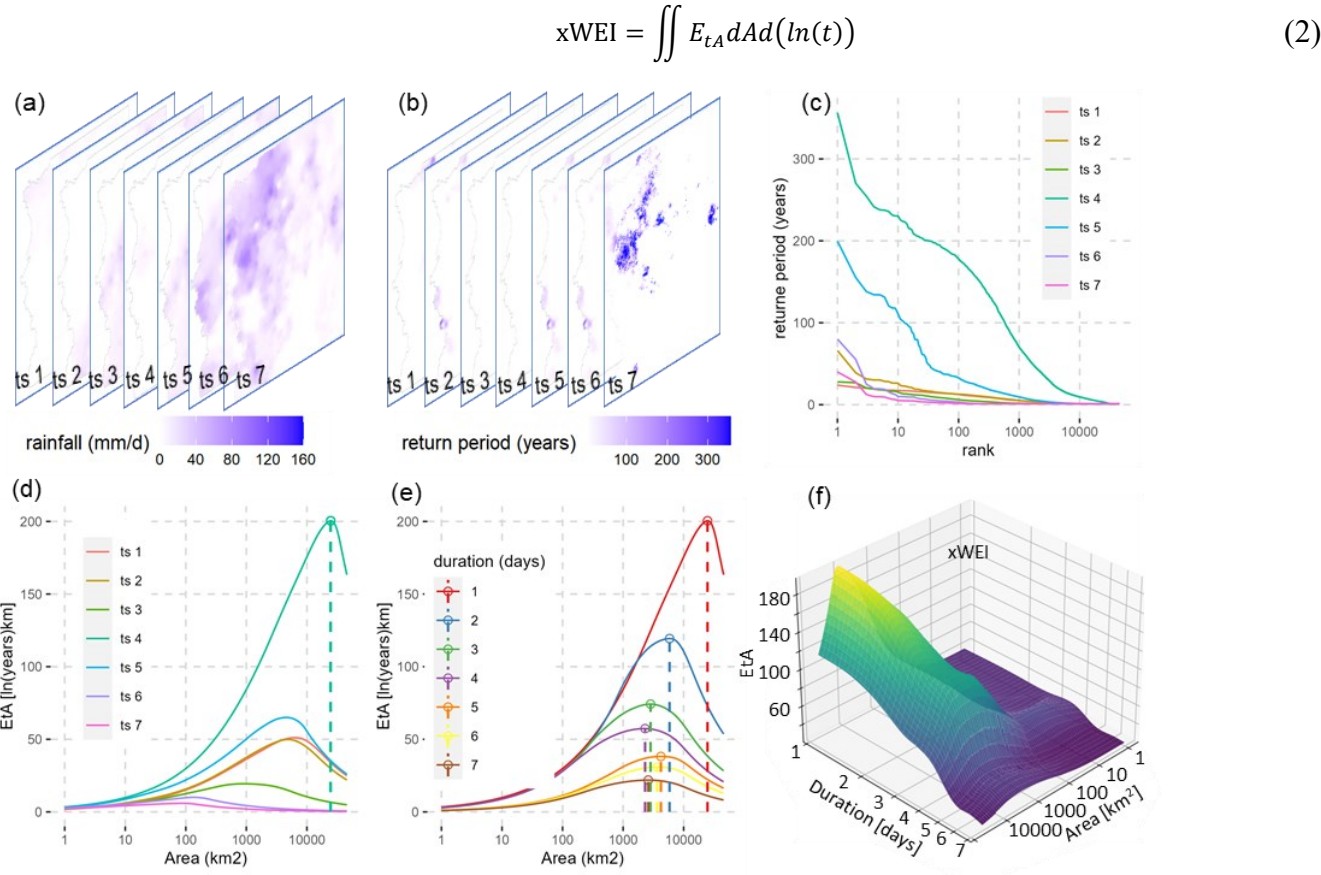


**Figure 2. Calculation of the WEI and xWEI. (a) maps of daily rainfall for a precipitation event lasting 7 days; (b) return period at each individual grid cell of each map; (c) sorting of the return period estimates for each duration in decreasing order for each map; (d) calculation of the $E_{tA}$ curves and selection of the curve with the highest peak to represent the extremity pattern of the precipitation for the duration of 1 day; (e) repetition of the procedures (a-d) for other durations (2, 3, …, 7 days) to derive the $E_{tA}$**
**curves; (f) the $E_{tA}$ curves from (e) are placed on a grid and a 3D-surfaceof $E_{tA}$ is interpolated; xWEI is defined as the volume underneath this surface.**

To estimate the return periods required to obtain $E_{tA}$, we use the Generalized Extreme Value (GEV) distribution, which has been found suitable for modeling precipitation extremes (Fowler and Kilsby, 2003) and has also been used in previous WEI studies (Gvoždíková et al., 2019; Minářová et al., 2018; Müller and Kaspar, 2014). The computation of the xWEI requires





return periods for multiple durations (1 to 7 days in our case). We use the dGEV method (Koutsoyiannis et al., 1998) to estimate return periods consistently across durations for each grid cell. Previous studies have shown that this method preserves the tail behaviour of precipitation extremes across durations and reduces the uncertainty (Ulrich et al., 2020; Fauer et al., 2021). The parameters of the dGEV distribution are obtained by Maximum Likelihood Estimation using the R package IDF (Ulrich et al., 2021).

To assess the extremity of HPEs across Germany, durations from 1 to 7 days are selected. While the WEI and xWEI can be extended to sub-daily scales (Voit and Heistermann, 2022), sub-daily precipitation observations are generally scarce and cover only recent decades, and most weather generators are set up at the daily scale. Furthermore, short-duration high-intensity precipitation events usually affect comparatively small areas and HPEs spanning several days involve large areas (Lengfeld et al., 2021; Lengfeld et al., 2019; Orlanski, 1975). Durations of up to 7 days are considered long enough to cover

HPEs that cause disastrous flood damage in Germany (Ganguli and Merz, 2019).

The most extreme HPEs for these 7 durations across Germany are identified by extracting annual maximum WEI values for each duration based on E-OBS precipitation data for the historical period 1950-2021. The corresponding WEI-areas (the area where the $E_{tA}$ curve reaches its peak WEI) of the annual most extreme HPEs for each duration are categorized into 6 classes (Figure 3) in order to analyze the spatial properties of HPEs in Germany. The same procedure is carried out for each of the

100 realizations to evaluate the performance of nsRWG in reproducing the cross-scale properties of HPEs in Germany.

## 4 Results and discussion

### 4.1 nsRWG performance for WEI

Figure 3 shows the frequency distribution of the observed (E-OBS) and simulated (nsRWG) WEI-areas of the annual maximum HPEs for 7 different durations. The distributions of the annual maximum WEI values are given in Figure 4. The

results show that more than half of the annual maximum HPEs in Germany are events with WEI-areas $\leq 20 \times 10^3$ km$^2$ for all durations. HPEs with larger spatial extent of $20 \times 10^3$ km$^2$ or more have a frequency of less than 20% in the past 72 years. However, these events are the most severe ones regarding their WEI values (Figure 4). The nsRWG is able to reproduce the annual maximum WEI and its corresponding areas for the 7 durations. The boxplots in Figures 3 and 4 show that the simulated distribution patterns are in good agreement with the E-OBS observations. However, the frequencies of HPEs with

WEI-areas $\leq 20 \times 10^3$ km$^2$ for the shorter durations (1d and 2d) are overestimated, while the frequencies of HPEs with larger WEI-areas are underestimated (Figure 3). For durations of 3 days and longer, the observed and simulated HPE frequencies are more balanced.
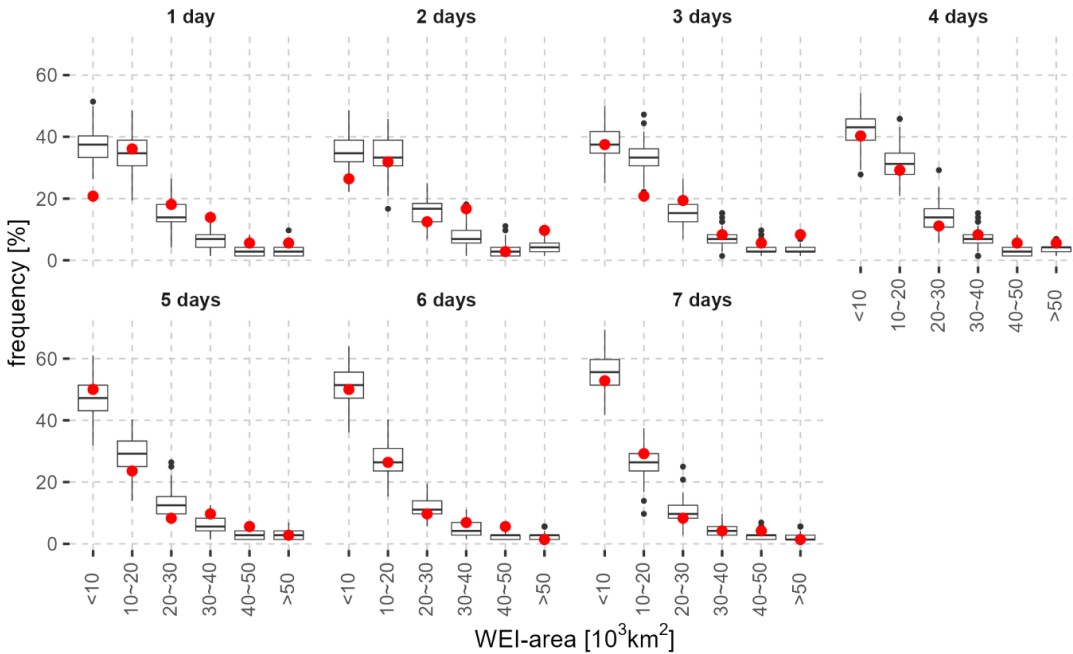

**Figure 3: Frequency distribution of observed and nsRWG-simulated WEI-areas of the annual maximum HPEs in the period 1950-2021 for 7 durations. The titles of the subplots are the durations of interest. The red dots denote the frequency of HPEs in different area categories from E-OBS precipitation data and the boxplots represent the simulated frequency distribution from 100 nsRWG realizations.**

The relation between the annual maximum WEI values and their return periods derived from the nsRWG data agrees well with the E-OBS observations for the 7 durations (Figure 5). This comparison confirms the ability of the nsRWG to simulate the occurrence probability of HPEs in Germany. The observed probability plots (red dots in Figure 5) are well enclosed by the simulated ranges (shadowed areas) from the nsRWG realizations. This is especially true for high return periods, which demonstrates the good performance in simulating HPEs for different durations. For return periods between 2 and about 10 years and short durations, nsRWG slightly underestimates the observed WEI. In contrast to traditional intensity-duration-frequency curves, there is no systematic difference between the empirical probability plots of annual maximum WEI series of different durations, as the WEI is based on return periods rather than precipitation totals or averages. This makes the WEI index comparable across temporal scales.




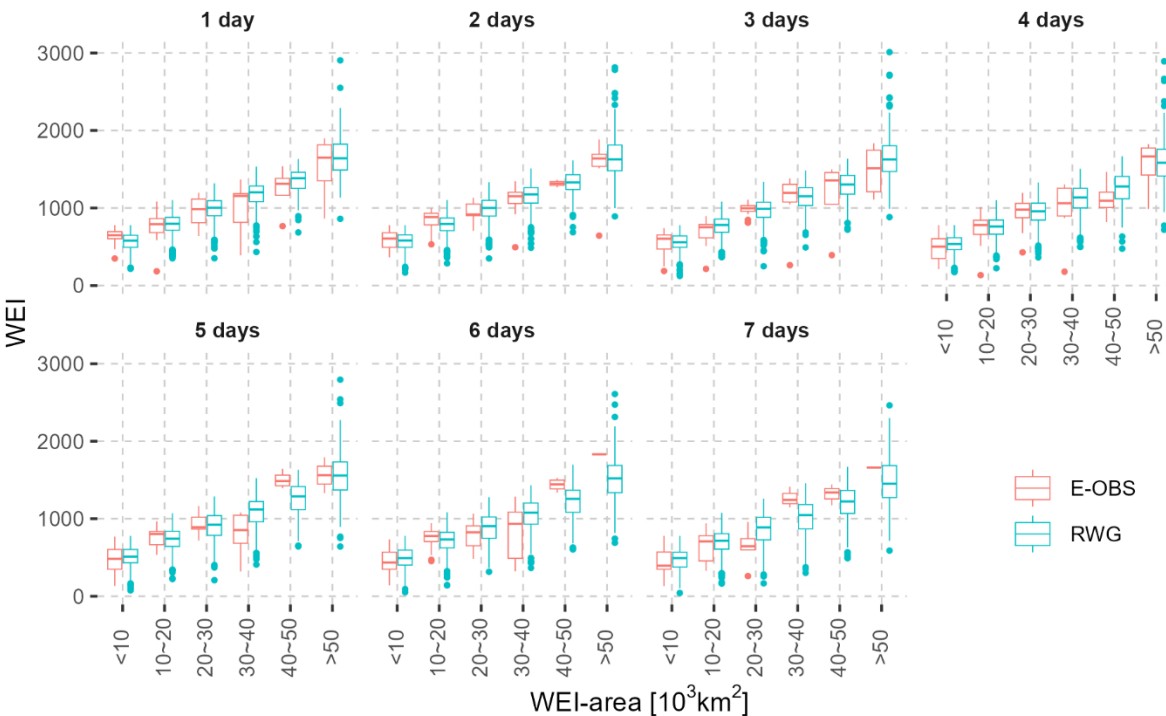

Figure 4: Distribution patterns of observed (red) and nsRWG-simulated (blue) annual maximum WEI series (denoted in y-axis, with the same unit as $E_{tA}$: ln(years)·km) for 7 durations. The WEI-areas of HPEs are categorized into 6 classes (see x-axis).




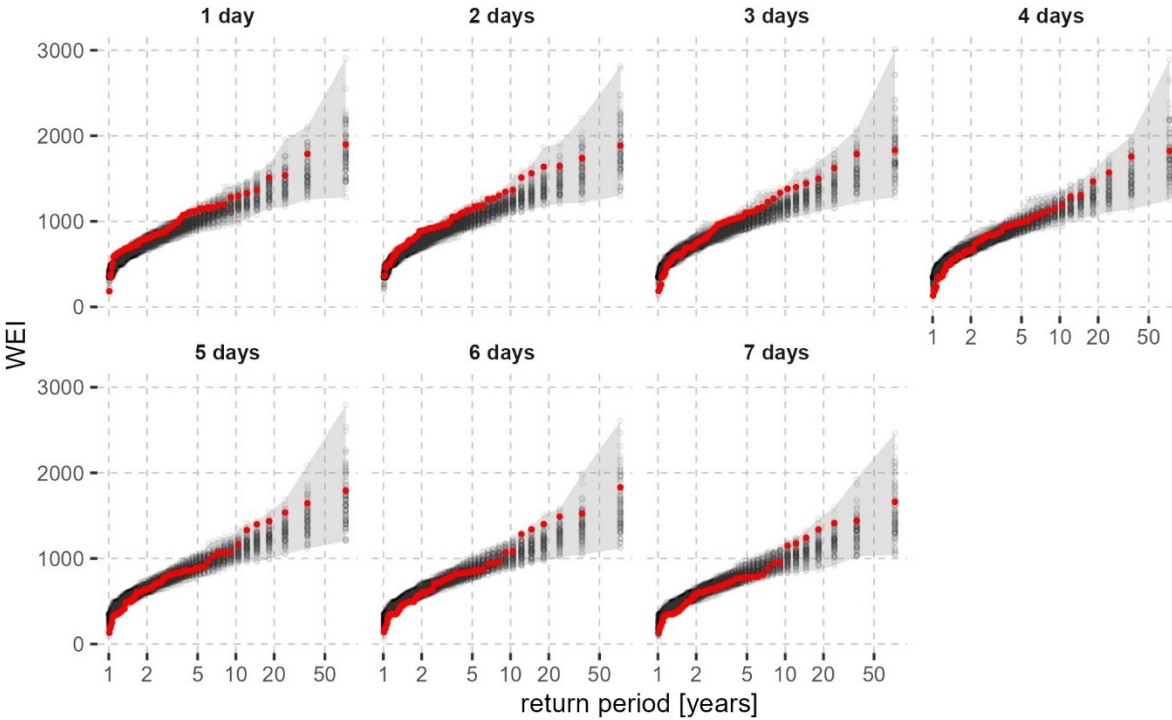

**Figure 5: Empirical probability plots of E-OBS observed (red dots) and nsRWG simulated (black dots) annual maximum WEI for**
**7 durations. The Weibull plotting positions (Makkonen, 2006) are used to estimate the return periods. The grey shaded area**
**indicates the upper and lower boundary of the nsRWG simulations.**

### 4.2 nsRWG performance for xWEI

We compute the xWEI for HPEs in Germany for the period 1950-2021 and extract the annual maximum series from both the
E-OBS dataset and the nsRWG realizations. The empirical probability plots of annual maximum xWEI, based on Weibull
plotting positions, agree well for both datasets: the cross-scale extremity index xWEI of the observed data lies within the
range of the 100 realizations (Figure 6). However, for return periods between 2 and 5 years, the realizations of nsRWG tend
to underestimate the xWEI, similar to the performance with respect to the WEI.

Figure 7 shows the extremity pattern of a real HPE event in August 2002 – one of the most damaging events in Germany. In
addition, HPEs from nsRWG realizations with similar xWEI values to the August 2002 HPE are shown. Figure 7 (b)
demonstrates that the nsRWG is able to generate HPEs with spatial and temporal extremity patterns very similar to the
August 2002 event, characterized by the highest extremity for the durations of 1d and 2d and an affected area of
approximately 20 000 km². The other two nsRWG-generated HPEs with similar xWEI values as the August 2002 event
show different cross-scale extremity patterns (emphasis on longer durations of 3-4 days). The reproducibility of historical
HPEs illustrates the ability of nsRWG in representing the cross-scale extremity of HPEs in Germany. The variations in

synthetic events and the respective $E_{tA}$ surfaces further demonstrate how nsRWG can generate synthetic events, which are

similar in terms of their cross-scale extremity but have their emphasis at different spatial and temporal scales.

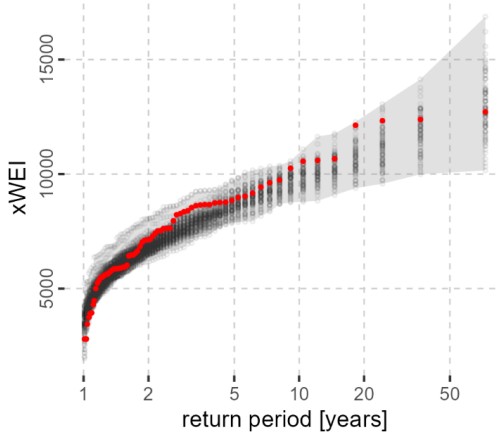

**Figure 6: Empirical probability plots of observed (E-OBS, red dots) and simulated (nsRWG, black dots) annual maximum xWEI. The Weibull plotting position is used to estimate the empirical return periods. The grey shadowed area indicates the upper and 230    lower boundary of simulated xWEI from 100 nsRWG realizations.**

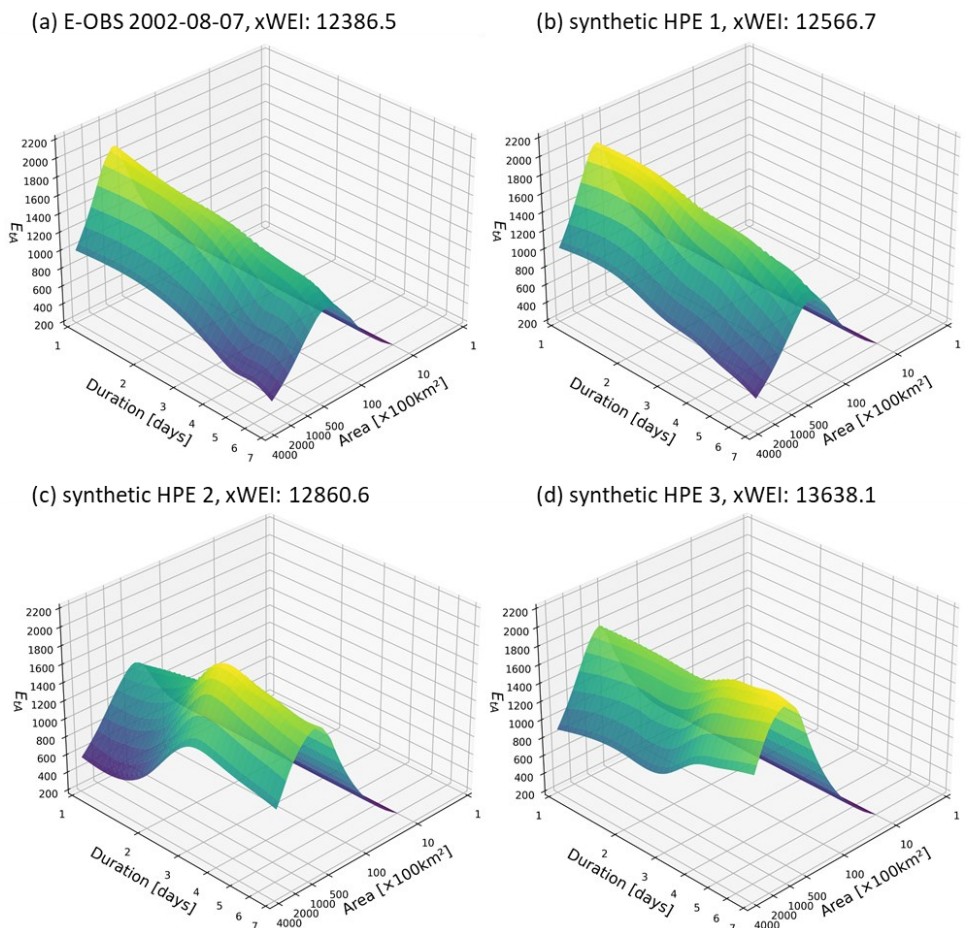

**Figure 7: Cross-scale extremity pattern (interpolated $E_{tA}$ curves over duration and area) of HPEs for (a) the August 2002 event and (b-d) 3 HPEs with similar xWEI values generated by nsRWG. The HPE in (b) shows great similarity in the cross-scale extremity pattern with the August 2002 event in (a), while the HPEs in (c) and (d) display different extremity patterns, although their xWEI values are similar to the actual event in August 2002.**

## 5 Conclusions

In this study, we evaluated the ability of a multi-site stochastic regional weather generator to capture the extremity of HPEs across spatial and temporal scales. For this purpose, we set up the nsRWG at a large scale (covering all of Germany and riparian regions) and generated 100 realizations of 72 years of synthetic precipitation data at daily resolution. The performance evaluation of nsRWG in simulating precipitation focuses on the event scale and extreme cases. This focus complements typical proxy statistics (like mean and standard deviation) that tend to only represent the general properties of WG in precipitation generation. Two indices, WEI and xWEI, are used to measure the extremity of observed and synthetic HPEs, both of them based on the spatial aggregation of return periods of precipitation totals for several durations of interest.





While WEI quantifies the maximum extremity of an event that occurred at a specific spatial extent and temporal duration, xWEI integrates extremity across the spatial and temporal scales of interest. The results demonstrate that nsRWG performs well in simulating the extremity patterns across most spatial and temporal scales of HPEs in Germany. However, it tends to overestimate the frequency of events with short durations and relatively small spatial extents. Using the August 2002 event as an example, we illustrate how the nsRWG is able to generate precipitation events with spatio-temporal extremity patterns

similar to those of historical disaster-causing HPEs.

With regard to future research, we emphasize that the choice of the spatial domain at which WGs are evaluated (here: all of Germany) is always a trade-off: on the one hand, the impacts of HPEs unfold at the catchment scale, and it would be an obvious next step to evaluate the ability of the WG to reproduce the frequencies of WEI and xWEI occurrence at the scale of specific river catchments (at the cost of computational effort, as this would require a higher spatial resolution). On the other

hand, we need to be aware that an HPE that occurred in one catchment might as well occur in an adjacent catchment, so that an analysis at a larger spatial domain (as the present one) is certainly warranted. Such considerations also show the links to "spatial counterfactuals", which have recently gained attention (Merz et al., 2023, Voit and Heistermann, 2024, Vorogushyn et al., 2024). Using spatial counterfactual scenarios, we can investigate the impact of HPEs in the hypothetical case that they had happened elsewhere. Weather generators could be a useful tool to explore how events with similar extremity indices

could unfold in different locations of the spatial domain, or with different spatio-temporal signatures, hence supporting counterfactual research in evaluating the plausibility of counterfactual scenarios.

**Code/Data availability**

The code and data to exemplify the computation of both WEI and xWEI can be found in the following repository:

https://doi.org/10.5281/zenodo.6556463 (Voit, 2022). The gridded precipitation data from E-OBS (version 25.0e; Cornes et al., 2018) is available at the European Climate Assessment & Dataset (ECA&D, https://www.ecad.eu/download/ensembles/download.php). The ERA5 mean sea level pressure and daily air temperature at 2 m height covering Europe can be found at https://cds.climate.copernicus.eu/cdsapp#!/dataset/reanalysis-era5-single-levels?tab=overview.


**Author contributions**

XG: data curation, conceptualisation, methodology, formal analysis, visualization, writing (original draft preparation). DVN: data curation, methodology, software. PV: methodology, software, writing – review & editing. BM: supervision, writing – review & editing. MK: writing – review & editing. SV: conceptualisation, supervision, writing (original draft preparation).


## Competing interests

The authors declare that they have no conflict of interest.

## Acknowledgments

This research has been funded by the Federal Ministry of Education and Research of Germany in the framework of the project FLOOD (project number 01LP2324E) as a part of the ClimXtreme Research Network on Climate Change and Extreme Events within the framework program Research for Sustainable Development (FONA3). Xiaoxiang Guan is funded by the China Scholarship Council for his PhD research (Grant #: 202106710029). We acknowledge the E-OBS dataset from the EU-FP6 project UERRA (http://www.uerra.eu) and the Copernicus Climate Change Service, and the data providers in the ECA&D project (https://www.ecad.eu).

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
