# Peer review of "The ability of a stochastic regional weather generator to reproduce heavy precipitation events across scales"

_Natural Hazards and Earth System Sciences, 2024_

## Referee Comment (RC1)

REVIEW of « The ability of a stochastic regional weather generator to reproduce heavy precipitation events across scales », by X Guan et al.

This paper presents the application of two statistical methods to evaluate how a recently updated « non stationary Regional Weather Generator » (nsRWG) produces heavy rain events over Germany.
The manuscript well explains the main features of the nsRWG and describes exhaustively the statistical methods used for the evaluation. I recommend publication in NHESS once the minor comments below have been solved.

GENERAL COMMENT
It would be highly beneficial for non expert readers to give more references useful for the physical interpretation of values of WEI and xWEI, i.e., examples of how the values of WEI and xWEI link to extreme event extension and duration. This can be added/extended in the Methods and Results and Discussion Section (although in section 4.2 an example is already mentioned).

TECHNICAL COMMENTS
Lines 53-55 : Merge or connect sentences – it is not evident that they are discussing the same issue.
Lines 119-120 : Specify that « SANDRA » is used to classify / cluster the circulation patterns.
Lines 129-130 : Specify that WEI is computed for each HPE !
Lines 133-134 : Clarify that $E\_tA$ is computed once for every value of A. Moreover, the dependency of A on n should be expressed in Equation 1.
Lines 165-170 : In my opinion this paragraph, explaining why sub-daily precipitation is not considered, can be easily compressed as a note of a couple of lines, since it is a limitation, and not relevant for the discussion of your results.
Lines 185-187 : The last two sentences are not coherent with each other. Please adjust.
Lines 197-198 : observed WEI is underestimated also for duration > 4 days and return periods between 10 and 20 years.
Line 200 : Is it worth mentioning here that the return period increases with duration, when considering fixed WEI values?
Lines 219-222 : How are the synthetic events in the comparison selected? Is there a way of selecting these based on the highest similarity in the WEI surface profile (i.e., sampling similar events in terms of extent and duration) ?
Line 248 : Doesn't it underestimate the frequency, since the return periods for the same WEI value are higher in the synthetic data?
Line 257 : Explain briefly what spatial counterfactuals are.

---

## Author Response (AR1)

**Response to reviewer 1**

REVIEW of « The ability of a stochastic regional weather generator to reproduce heavy precipitation events across scales », by X Guan et al.

This paper presents the application of two statistical methods to evaluate how a recently updated « non stationary Regional Weather Generator » (nsRWG) produces heavy rain events over Germany. The manuscript well explains the main features of the nsRWG and describes exhaustively the statistical methods used for the evaluation. I recommend publication in NHESS once the minor comments below have been solved.

GENERAL COMMENT

It would be highly beneficial for non-expert readers to give more references useful for the physical interpretation of values of WEI and xWEI, i.e., examples of how the values of WEI and xWEI link to extreme event extension and duration. This can be added/extended in the Methods and Results and Discussion Section (although in section 4.2 an example is already mentioned).

Reply: Thank you for the valuable suggestion. In the revised manuscript, we expanded the interpretation of Figure 2 to provide more clarity on the relationship between WEI, xWEI, and extreme event characteristics. Specifically, we explain how the extent and duration relate to the extremity pattern of the HPE event illustrated in Figure 2:

"For the exemplary HPE event, the analysis highlights that the daily extremity (in terms of 1-day rainfall intensity) occurred on the 4th day, with the highest EtA curve among the 7 days (Figure 2d). Comparing extremity across durations, the HPE was most extreme for a 1-day duration, affecting an area of over 10,000 km² (WEI-area). For longer durations (≥3 days), the HPE consistently influenced approximately the same area, showing a stabilization in spatial extent for these durations."

We believe this explanation offers a clearer physical interpretation of WEI and xWEI values and how they link to the extension and duration of extreme events.

TECHNICAL COMMENTS

Lines 53-55: Merge or connect sentences – it is not evident that they are discussing the same issue.

Reply: we follow reviewer's suggestion.

Lines 119-120: Specify that « SANDRA » is used to classify / cluster the circulation patterns.

Reply: thanks, "circulation pattern classification" is explicitly specified in SANDRA method description.

Lines 129-130: Specify that WEI is computed for each HPE

Reply: The sentence is rephrased as "The computation of WEI for individual HPE is illustrated in Figure 2."

Lines 133-134: Clarify that E_tA is computed once for every value of A. Moreover, the dependency of A on n should be expressed in Equation 1.

Reply: Thank you for pointing this out. The dependency of $E_{tA}$ on A has been explicitly clarified in the description of the $E_{tA}$ computation. Additionally, the equation in Line 156 has been revised to specify $A = \text{grid size} \times n$, ensuring clarity regarding the dependency of A on n.

Lines 165-170: In my opinion this paragraph, explaining why sub-daily precipitation is not considered, can be easily compressed as a note of a couple of lines, since it is a limitation, and not relevant for the discussion of your results.

Reply: Thank you for your comment. We have revised the paragraph to streamline the explanation and focus on the rationale for selecting durations from 1 to 7 days. The updated

version emphasizes that these durations are sufficient to capture the events responsible for disastrous flood damage in Germany and briefly mentions the limitations of sub-daily precipitation observations. This modification addresses your suggestion by making the paragraph more concise while retaining the necessary context for understanding the analysis.

Lines 185-187: The last two sentences are not coherent with each other. Please adjust.

Reply: The two sentences have been merged to improve coherence as requested.

Lines 197-198: observed WEI is underestimated also for duration > 4 days and return periods between 10 and 20 years.

Reply: we merged the two sentences "For return periods between 2 and about 10 years and short durations, nsRWG slightly underestimates the observed WEI" is modified to "For short durations and return periods between 2 and about 10 years, as well as for durations longer than 4 days with return periods between 10 and 20 years, the nsRWG slightly underestimates the observed WEI."

Line 200: Is it worth mentioning here that the return period increases with duration, when considering fixed WEI values?

Reply: It is generally correct that for HPEs with the same WEI value, the return period increases with duration. This indicates that the occurrence probability of an HPE with the same extremity (in terms of WEI magnitude) becomes less frequent as the duration increases. This relationship reflects the rarity of longer-duration extreme events with comparable intensity.

Lines 219-222: How are the synthetic events in the comparison selected? Is there a way of selecting these based on the highest similarity in the WEI surface profile (i.e., sampling similar events in terms of extent and duration)?

Reply: The selection of synthetic events with similar xWEI values to the 2021 event was conducted as follows: we first extracted HPEs with annual maximum xWEI from the 100 precipitation realizations generated by nsRWG. These synthetic events were then compared to the 2021 event based on their xWEI values. We acknowledge that only a few synthetic HPEs exhibited extremities close to the 2021 event. The three selected synthetic HPEs were chosen based on their xWEI surface characteristics and extremity across various durations. While quantifiable indicators, such as mean squared error (MSE), could provide an explicit measure of similarity in xWEI surface profiles, we decided not to include this detail in the manuscript as it does not significantly impact the core analysis.

Line 248: Doesn't it underestimate the frequency, since the return periods for the same WEI value are higher in the synthetic data?

Reply: Thank you for your comment. We believe there may be a misunderstanding. The statement in the conclusion "However, it tends to overestimate the frequency of events with short durations and relatively small spatial extents," refers to results shown in Figure 4, which depicts the distribution of WEI-area across areas and durations. Specifically, the nsRWG underestimates the frequency of events with short durations (1 day) and small spatial extents (<10,000 km²). In contrast, the statement about return periods for the same WEI value being higher in synthetic data reflects an inference from Figure 5. These address different aspects, and we therefore propose to keep the sentence unchanged.

Line 257: Explain briefly what spatial counterfactuals are.

Reply: Thank you for your comment. In the updated manuscript, we have included a general explanation of counterfactuals in the context of climate extremes: "Counterfactuals are scenarios that describe alternative ways an event could have unfolded (Woo, 2019; Montanari et al., 2024). These scenarios may involve altering or removing specific factors, such as anthropogenic climate change, natural climate variability, or other boundary conditions (Gauch et al., 2020)." Additionally, the manuscript already includes a description

of spatial counterfactuals: "Using spatial counterfactual scenarios, we can investigate the impact of HPEs in the hypothetical case that they had happened elsewhere." This provides the necessary context for understanding both counterfactuals and their spatial applications.

References:

Gauch, M. and Klotz, D. and Kratzert, F. and Nearing, S. and Hochreiter, S. and Lin, J. A Machine Learner's Guide to Streamflow Prediction. Workshop on AI for Earth Sciences 34th Conference on Neural Information Processing Systems (NeurIPS 2020) Vancouver, Canada, 2020.

Montanari, A., Merz, B., & Blöschl, G. (2024). HESS Opinions: The sword of Damocles of the impossible flood. *Hydrology and Earth System Sciences*, *28*(12), 2603–2615. https://doi.org/10.5194/hess-28-2603-2024

Woo, G. (2019). Downward Counterfactual Search for Extreme Events. *Frontiers in Earth Science*, *7*, 340. https://doi.org/10.3389/feart.2019.00340

**Response to reviewer 2**

This study presents a novel evaluation approach for WGs to examine their ability to capture cross-scale HPEs. The core method encompasses the WEI and xWEI, where the former focuses on the extremity of events at a single spatiotemporal scale, while the latter emphasizes the overall extreme performance of events across multiple scales. Their results indicate that nsRWG is capable of reproducing most HPEs in terms of duration (1 to 7 days), especially those that historically triggered disasters. However, nsRWG slightly overestimates the frequency of short-term (1-2 days) HPEs with smaller WEI areas. The manuscript was good in general, but I have some questions and comments to further improve the quality.

1. The introduction only briefly mentions Stochastic weather generators (WGs), and then directly jumps to Section 3.1, which discusses the version of the stochastic multi-site non-stationary regional weather generator (nsRWG) used in this study. It would be better to explain the similarities and differences between the two systems for the general readers.

Reply: Thank you for your suggestion. In the revised manuscript, we have added a definition, some typical characteristics, and the potential applications of stochastic weather generators (WGs) to the introduction section.

*"WGs are stochastic models that are capable to generate synthetic spatio-temporal fields of weather variables such as precipitation, temperature, humidity etc., retaining statistical properties of observed or climate model data on which WG is conditioned, such as autocorrelation, spatial covariance and multi-variable dependence. A large number of WG models have been introduced so far, based on various statistical methods among others reshuffling and perturbing analogue weather fields or applying multi-variate auto-regressive models (for a review, see Maraun et al., 2010, Haberlandt et al., 2011, Serinaldi and Kilsby, 2014, Benoit and Mariethoz, 2017, Nguyen et al., 2021). WGs can be instrumental in generating synthetic HPEs, thereby supporting flood risk management and climate adaptation (Breinl et al., 2013; Chen and Brissette, 2014; Harris et al., 2014, Sairam et al., 2021). WGs are widely used for estimation of hydrological design values (Winter et al., 2019),*

*downscaling climate model output (Fatichi et al., 2011, Kiem et al., 2021), climate impact assessments on water resources (Harris et al., 2014, Najibi et al., 2024), and flood risk assessment (Sairam et al., 2021) providing long-term datasets for scenarios where observational data may be limited and downscaled future climate projections are needed. WGs are particularly effective when integrated with other models to better understand and prepare for HPEs and their consequences (Mehrotra and Sharma, 2010; Zhou et al., 2020)."*

This addition provides a clearer context for readers unfamiliar with WGs. Furthermore, in Section 3.1, we have simplified the expressions by directly introducing the structure of the nsRWG used in this study, omitting introductions of previous versions to maintain focus and avoid unnecessary details.

2. In Section 3.2, the explanation regarding how the return periods ($P_{(t,i)}$) used in the calculation process are obtained is somewhat unclear. Please provide further clarification.

Reply: Thank you for this comment. In the revised manuscript, we have clarified the procedures for return period estimation. Specifically, the duration-dependent GEV distribution is employed to derive the intensity-duration-frequency (IDF) relationship for precipitation at each grid cell. This IDF relationship is then used to estimate the return period of rainfall intensity at each grid cell, which subsequently informs the EtA computation.

3. This study only uses the cross-scale weather extremity index (xWEI) as an assessment standard to compare and discuss the outputs of E-OBS (observational data) and nsRWG (simulated data). Both xWEI and the nsRWG methods were referred to other studies (this study only combined two systems). Given the amount of content for publication, it might be not enough. Please find a way to extend the current analysis or results based on the research objectives. For example: (1) Incorporate other models as a control group to evaluate the advantages of nsRWG compared to other technologies; (2) Use traditional

validation scores or provide additional reference to illustrate the differences in Figure 7 for various cases; (3) Further explain the relevant statistics constructed in this paper and their practical applications in the prevention and control of Heavy Precipitation Events (HPEs).

Reply: Thank you for your insightful suggestions. Regarding the evaluation of the nsRWG model, we would like to highlight that it has been thoroughly validated in previous studies using a traditional evaluation framework. Specifically, Nguyen et al. (2024) rigorously assessed the performance of the nsRWG model across multiple statistical metrics, including precipitation intermittency, wet/dry transition probabilities, intensities of high percentiles, spatial correlation of rainfall across grid cells, and catchment areal precipitation averages. These evaluations confirm the model's robustness and reliability in simulating key rainfall characteristics in Germany.

Given this comprehensive prior validation, we focused our study on the event-scale characteristics of heavy precipitation events (HPEs), specifically the spatio-temporal integrated extremity represented by WEI and xWEI. This approach aligns with our research objectives to analyze HPE extremity patterns rather than statistical properties of bulk precipitation or selected high percentile precipitation intensities. Thus, we did not include comparisons based on traditional metrics in this manuscript. We believe this focus provides novel insights into understanding HPEs at the event scale, complementing the evaluation study of the nsRWG model by Nguyen et al. (2024).

Thank you for pointing out the importance of linking the statistics in our study to practical applications. We have elaborated on the practical relevance of our findings in the revised manuscript. Specifically, our study highlights how integrated event-scale metrics like WEI and xWEI provide insights into the spatial and temporal extremity patterns of HPEs. These metrics are directly relevant for evaluating the potential impacts of extreme precipitation events and informing risk management strategies. In the discussion and conclusion, we connect our findings to their potential applications for studies using "spatial counterfactuals."

This approach can help explore hypothetical scenarios where HPEs unfold differently, such as occurring in neighboring catchments or under perturbed boundary conditions. Counterfactual scenarios provide valuable insights for disaster prevention and preparedness by illustrating alternative outcomes of extreme weather events. Weather generators, like the one used in our study, offer a way to simulate these scenarios, enabling an assessment of their plausibility and potential impacts.

References:

Benoit, L. and Mariethoz, G. Generating synthetic rainfall with geostatistical simulations. Wiley Interdisciplinary Reviews-Water, 4(2), doi:10.1002/wat2.1199, 2017.

Breinl, K., Turkington, T. and Stowasser, M. Stochastic generation of multi-site daily precipitation for applications in risk management. Journal of Hydrology, 498, 23-35, doi:10.1016/j.jhydrol.2013.06.015, 2013.

Chen, J. and Brissette, F.P. Stochastic generation of daily precipitation amounts: review and evaluation of different models. Climate Research, 59(3), 189-U145, doi:10.3354/cr01214, 2014.

Fatichi, S., V. Y. Ivanov, and E. Caporali, 2011: Simulation of future climate scenarios with a weather generator. Adv. Water Resour., 34, 448–467, https://doi.org/10.1016/j.advwatres.2010.12.013.

Haberlandt, U., Hundecha, Y., Pahlow, M., Schumann, A.H. Rainfall generators for application in flood studies. In: Schumann, A.H. (Ed.), Flood Risk Assessment and Management. Springer, Netherlands, pp. 117-147, doi:10.1007/978-90-481-9917-4_7, 2011.

Harris, C.N.P., Quinn, A.D. and Bridgeman, J. The use of probabilistic weather generator information for climate change adaptation in the UK water sector. Meteorological Applications, 21(2), 129-140, doi:10.1002/met.1335, 2014.

Kiem, A. S., Kuczera, G., Kozarovski, P., Zhang, L., & Willgoose, G. (2021). Stochastic generation of future hydroclimate using temperature as a climate change covariate. Water Resources Research, 57, 2020WR027331. https://doi.org/10.1029/2020WR027331

Maraun, D., Wetterhall, F., Ireson, A.M., Chandler, R.E., Kendon, E.J., Widmann, M., Brienen, S., Rust, H.W., Sauter, T., Themel, M., Venema, V.K.C., Chun, K.P., Goodess, C.M., Jones, R.G., Onof, C., Vrac, M. and Thiele-Eich, I. Precipitation downscaling under climate change: Recent developments to bridge the gap between dynamical models and the end user. Reviews of Geophysics, 48(3), doi:10.1029/2009RG000314, 2010.

Mehrotra, R. and Sharma, A. Development and Application of a Multisite Rainfall Stochastic Downscaling Framework for Climate Change Impact Assessment. Water Resources Research, 46(7), doi:10.1029/2009WR008423, 2010.

Najibi, N., Perez, A.J., Arnold, W., Schwarz, A., Maendly, R. and Steinschneider, S. 2024. A statewide, weather-regime based stochastic weather generator for process-based bottom-up climate

risk assessments in California – Part II: Thermodynamic and dynamic climate change scenarios. Climate Services, 34, 100485. doi: https://doi.org/10.1016/j.cliser.2024.100485.

Nguyen, V.D., Merz, B., Hundecha, Y., Haberlandt, U. and Vorogushyn, S. Comprehensive evaluation of an improved large-scale multi-site weather generator for Germany. International Journal of Climatology, 41(10), 4933-4956, doi:10.1002/joc.7107, 2021.

Nguyen, V. D., Vorogushyn, S., Nissen, K., Brunner, L., & Merz, B. (2024). A non-stationary climate-informed weather generator for assessing future flood risks. Advances in Statistical Climatology, Meteorology and Oceanography, 10(2), 195–216. https://doi.org/10.5194/ascmo-10-195-2024

Sairam, N., Brill, F., Sieg, T., Farrag, M., Kellermann, P., Nguyen, V.D., Lüdtke, S., Merz, B., Schröter, K., Vorogushyn, S. and Kreibich, H. Process-Based Flood Risk Assessment for Germany. Earth's Future, 9(10), e2021EF002259, doi:10.1029/2021EF002259, 2021.

Serinaldi, F. and Kilsby, C.G. Simulating daily rainfall fields over large areas for collective risk estimation. Journal of Hydrology, 512, 285-302, doi:10.1016/j.jhydrol.2014.02.043, 2014.

Winter, B., Schneeberger, K., Dung, N.V., Huttenlau, M., Achleitner, S., Stötter, J., Merz, B. and Vorogushyn, S. A continuous modelling approach for design flood estimation on sub-daily time scale. Hydrological Sciences Journal, 64(5), 539-554. doi: 10.1080/02626667.2019.1593419, 2019.

Zhou, L., Meng, Y., Lu, C., Yin, S. and Ren, D. A frequency-domain nonstationary multi-site rainfall generator for use in hydrological impact assessment. Journal of Hydrology, 585, doi:10.1016/j.jhydrol.2020.124770, 2020.